# Zero-Velocity Update-Based GNSS/IMU Tightly Coupled Algorithm with the Constraint of the Earth’s Rotation Angular Velocity for Cableway Bracket Deformation Monitoring

**DOI:** 10.3390/s23249862

**Published:** 2023-12-16

**Authors:** Song Zhang, Qiuzhao Zhang, Ruipeng Yu, Zhangjun Yu, Chu Zhang, Xinyue He

**Affiliations:** 1No. 1 Institute of Geology and Mineral Resources of Shandong Province, Jinan 250109, China or z2013196@sdaeu.edu.cn (S.Z.); qiuzhao.zhang@cumt.edu.cn (Q.Z.); 2School of Environmental and Spatial Informatics, China University of Mining and Technology, Xuzhou 221116, China; tb22160007a41@cumt.edu.cn (Z.Y.); ts20160184p31@cumt.edu.cn (C.Z.); ts23160007a31ld@cumt.edu.cn (X.H.)

**Keywords:** GNSS/IMU, tightly coupled, ZUPT, Earth’s rotation angular velocity, deformation monitoring

## Abstract

Cableways have been widely used in industrial areas, cities, and scenic spots due to their advantages, such as being a convenient mode of transportation, time-saving, labor-saving, and low cost, as well as offering environmental protection. To ensure the safe operation of a cableway, based on the characteristic that the velocity of the cableway bracket is approximately zero in a static deformation monitoring environment, a deformation monitoring method called zero velocity update (ZUPT)-based GNSS/IMU tightly coupled algorithm with the constraint of the Earth’s rotation angular velocity was proposed. The proposed method can effectively solve the problem of a single GNSS being unable to output attitude, which is directly related to the status of wire ropes and cable cars. Meanwhile, ZUPT is used to restrain the Kalman filter’s divergence when IMU is stationary. However, the improvements of ZUPT on attitude are not obvious, so the constraint of the Earth’s rotation angular velocity was applied. The performance of the proposed method was evaluated through monitoring the cableway bracket of the Yimeng Mountain Tourism area in Shandong. Compared with the ZUPT-based GNSS/IMU tightly coupled algorithm (ZUPT-TC), the proposed method can further constrain the error accumulation of IMU while stationary and, therefore, it can provide reliable position and attitude information on cableway brackets.

## 1. Introduction

Cableways are transportation tools that have functions such as freight and passenger transportation. Compared with other transportation equipment, a cableway has the advantages of being a convenient mode of transportation, time-saving, labor-saving, and low cost, as well as offering environmental protection, and it is therefore widely used in tourism [1,2]. Nowadays, the number of in-use passenger cableways in China has reached 1148, and more than 4000 patents for passenger cableways have been disclosed. However, due to the relatively late development and incomplete technology of cableway safety monitoring in China, accidents related to cableways are constantly occurring in the Yunnan Xishuangbanna Cableway, Hunan Tianmenshan Cableway, Shanxi Huashan Cableway, Nanjing Zijinshan Cableway, and other areas. In this case, safety monitoring technology is a must to ensure the safe operation of the cableway [3]. Generally, the support device of the cableway is composed of a steel wire rope and a cableway bracket. It is acknowledged that the occurrence of cableway accidents is often due to damage to the steel wire rope, and the cableway bracket is used to support the normal operation of the steel wire rope, so potential faults and abnormal states of the cableway can be detected in a timely manner by monitoring the three-dimensional deformation and attitude changes in the cableway bracket [4]. Therefore, it is imperative to monitor the deformation and attitude changes in the cableway bracket.

Periodic contact measurement is a common method used in monitoring the deformation and attitude changes in cableway brackets, but it has poor timeliness and low efficiency. In addition, periodic contact measurement cannot reflect the operational status of the cableway in real time. In order to improve the accuracy and real-time performance of monitoring, dual- or multi-antenna GNSSs can be used to obtain the deformation and attitude changes in the cableway bracket [5,6]. Bock proposed that GPS can be used to monitor surface deformation [7]. Subsequently, GPS technology was applied to deformation monitoring of natural gas pipelines and achieved sub-centimeter-level monitoring accuracy [8]. In [9], GPS was found to be able to monitor building vibration frequencies of 1.2 Hz, and it was proven that the use of GNSS for deformation monitoring is actually feasible. What is more, GNSS was also successfully applied in bridge structural deformation and deflection monitoring [10], dam deformation monitoring [11], and volcanic turbulence and post-earthquake deformation monitoring [12]. Unfortunately, high-precision attitude is difficult to obtain directly through GNSS due to the structural defects in the design of the cableway bracket.

The inertial measurement unit (IMU) is based on Newton’s mechanical laws of relative inertial space, using inertial components such as gyroscopes and accelerometers to sense the acceleration and angular velocity of the carrier during motion. Then, the position, velocity, and attitude navigation information of the carrier can be obtained by integral calculation. The IMU has the advantages of strong independence, anti-interference ability, and high sampling frequency, and is widely used in integrated navigation [13,14,15]. However, extended time sampling of INS in a stationary state will lead to drifting error [16]. Therefore, the outstanding short-term relative measurement accuracy of INS in integrated navigation applications was studied and analyzed [17,18,19]. The GNSS/IMU integrated method can fully utilize the accuracy of GNSS positioning, as well as the high sampling frequency and attitude acquisition ability of IMU [20], so it can provide accurate monitoring data for cableway operation. In the environment of dynamic GNSS/IMU deformation monitoring, the feasibility of integrated navigation for track deformation monitoring has been verified, which means that integrated navigation can perform high-precision deformation monitoring [21,22,23]. Furthermore, road flatness measurement and track irregularity measurement can also be carried out through integrated navigation [24]. Different from dynamic measurement, GNSS/IMU static measurement has attracted great attention because of its unique properties in deformation monitoring. In [25], it was proven that integrated positioning can monitor high-frequency small deformation (such as seismic waves) and meet the accuracy requirements, which also provides theoretical preparation for the GNSS/IMU integrated monitoring of structural deformation. The evaluation and testing of the monitoring accuracy of the GNSS/IMU tightly coupled algorithm in complex environments using a vibration table in [26] also demonstrated that the GNSS/IMU combination can improve accuracy and robustness compared to GNSS alone. Aiming to address the problem of using fixed Earth gravity values and ignoring Earth’s rotation in traditional pre-integration algorithms, an IMU pre-integration algorithm considering Earth’s rotation and gravity changes was proposed, and it was shown to improve positioning accuracy by 32.41% and yaw accuracy by 4.23% [27]. Moreover, a GNSS/IMU robust adaptive filtering algorithm based on innovation was proposed, and it significantly improved positioning and attitude accuracy compared to extended Kalman filtering [28].

Currently, cableway deformation monitoring is mainly achieved through the establishment of an automated and information-based integrated platform for remote monitoring [29,30]. In [31], a monitoring system for passenger cableways was preliminarily designed that can monitor and diagnose cableway information. Afterward, a set of software for parallel cableway control and monitoring was developed and successfully applied [32]. With the further development of information technology, automatic, informationized, and networked monitoring has become the mainstream. Wireless networks and cloud services were used to improve the informatization level and monitoring efficiency of cableway deformation monitoring [33]. Meanwhile, cableway deformation monitoring based on multi-sensor fusion has generated wide interest, and there have been many studies on the monitoring of wire ropes in cableways. It was found that regular condition monitoring of steel wire ropes based on non-destructive investigation and evaluation techniques can improve safety and optimal rope life [34]. Lagerev et al. analyzed the wire rope of a cableway and observed the deformation of the wire rope through computer simulation of the overall time-varying reliability index [35]. Sukhorukov et al. used magnetic flux leakage testing (MFL) in automatic wire rope monitoring systems, which can perform automatic monitoring and save time and effort [36]. In addition, by combining GPR and Differential GPS (DGPS), high-frequency and high-precision data for cableway deformation monitoring can be obtained to establish a corresponding cableway deformation monitoring system [37]. In terms of maintenance, fuzzy inference systems can be used to simulate the failure modes of large-scale cableway transportation systems, thereby achieving maintenance automation [38]. In addition, there are relatively few studies conducted on deformation monitoring of other parts of the cableway. In [39], the safety detection device for the cableway bracket was thoroughly studied, and the fault monitoring process was successfully analyzed. Zhang et al. used acoustic emission technology to monitor the condition and diagnose faults of the bearings in the cableway [40].

Overall, current cableway deformation monitoring mainly focuses on the development of relevant automated monitoring platforms or research on the deformation monitoring of related parts such as cable cars and wire ropes. However, the cableway bracket used to support the normal operation of the steel wire rope has not been given enough attention, as potential faults and abnormal states of the cableway will have a significant impact on the three-dimensional deformation and attitude changes in the cableway bracket. Therefore, the goal of this paper is to propose a GNSS/IMU tightly coupled algorithm for cableway bracket deformation monitoring and accident avoidance. To overcome the problem of Kalman filter divergence and drifting error caused by static IMU measurement, ZUPT and the constraint of the Earth’s rotation angular velocity are applied to the measurement model and state model of the Kalman filter for the GNSS/IMU tightly coupled algorithm.

The remainder of this paper is organized as follows: Section 2 lists the formulas of the dynamic and observation equations of the GNSS/IMU tightly coupled algorithm and extended Kalman filter (EKF). In Section 3, the ZUPT and the constraint of the Earth’s rotation angular velocity are derived and described. The set of test results and data analysis are given in Section 4. Finally, the paper is concluded in Section 5.

## 2. Extended Kalman Filter for GNSS/IMU Tightly Coupled Algorithm

The EKF is widely considered as the best linear filter when the system state and measurement noise are both zero-mean Gaussian white noise. In this section, the dynamic and observation equations of the GNSS/IMU tightly coupled algorithm and the EKF are provided.

### 2.1. The Dynamic and Observation Equations of GNSS/INS Tightly Coupled Algorithm

The GNSS/INS tight combination is the process of inputting GNSS observations and predicted INS observations together into a Kalman filter, and obtaining the position and navigation results of the carrier through filtering fusion. Compared with loose combination, tight combination has better positioning accuracy and robustness, especially when the number of satellites is small or loss of lock occurs, it can significantly improve the availability of GNSS under harsh observation conditions. In addition, owing to the fact that tight combinations only need one filter for optimal estimation, they have higher computational efficiency in practical work. In this paper, the state model of the EKF based on the IMU error model for the GNSS/IMU tightly coupled system adopts an 18-dimensional state:(1)$$X={\left[\delta \psi^{e},\delta v^{e},\delta r^{e},b_{a}^{b},b_{g}^{b},dt_{r},d{\dot{t}}_{r},\nabla \Delta N\right]}^{T}$$
where X is the state error vector of GNSS/INS integrated navigation; $\delta \psi^{e}$, $\delta v^{e}$, and $\delta r^{e}$ represent the three-dimensional attitude error, velocity error, and position error, respectively; $b_{a}^{b}$ and $b_{g}^{b}$ are acceleration deviation and gyro drift in the carrier coordinate frame, respectively; $dt_{r}$ is the receiver clock error; $d{\dot{t}}_{r}$ is the receiver clock error drift; and ∇ΔN is the double difference ambiguity.

The dynamic model of the GNSS/INS tightly coupled navigation system can be expressed as:(2)$$X_{k}=\Phi_{k/k-1}X_{k-1}+w_{k-1}$$
(3)$$\Phi_{k/k-1}=\left[\begin{array}{cccccccc} F_{11} & F_{12} & F_{13} & 0 & -C_{b}^{n} & 0 & 0 & 0\\ F_{21} & F_{22} & F_{23} & C_{b}^{n} & 0 & 0 & 0 & 0\\ F_{31} & F_{32} & 0 & 0 & 0 & 0 & 0 & 0\\ 0 & 0 & 0 & F_{44} & 0 & 0 & 0 & 0\\ 0 & 0 & 0 & 0 & F_{55} & 0 & 0 & 0\\ 0 & 0 & 0 & 0 & 0 & 0 & 1 & 0\\ 0 & 0 & 0 & 0 & 0 & 0 & -t_{\alpha } & 0\\ 0 & 0 & 0 & 0 & 0 & 0 & 0 & 0 \end{array}\right]$$
$$F_{11}=\left[\begin{array}{ccc} 0 & 0 & 0\\ -\omega_{ie}\sin L & 0 & 0\\ \omega_{ie}\cos L+\frac{V_{e}}{R_{N}+H}\sec^{2}L & 0 & 0 \end{array}\right]$$
$$F_{12}=\left[\begin{array}{ccc} 0 & -\frac{1}{R_{M}+H} & 0\\ \frac{1}{R_{N}+H} & 0 & 0\\ \frac{\tan L}{R_{N}+H} & 0 & 0 \end{array}\right]$$
$$F_{13}=\left[\begin{array}{ccc} 0 & \omega_{ie}\sin L+\frac{V_{e}\tan L}{R_{N}+H} & -(\omega_{ie}\cos L+\frac{V_{e}}{R_{N}+H})\\ -(\omega_{ie}\sin L+\frac{V_{e}\tan L}{R_{N}+H}) & 0 & -\frac{V_{n}}{R_{M}+H}\\ \omega_{ie}\cos L+\frac{V_{e}}{R_{N}+H} & \frac{V_{n}}{R_{M}+H} & 0 \end{array}\right]$$
$$F_{21}=\left[\begin{array}{ccc} 2\omega_{ie}V_{n}\cos L+\frac{V_{e}V_{n}}{R_{N}+H}\sec^{2}L+2\omega_{ie}V_{u}\sin L & 0 & 0\\ -(2\omega_{ie}V_{e}\cos L+\frac{V_{e}^{2}}{R_{N}+H}\sec^{2}L) & 0 & 0\\ -2\omega_{ie}V_{e}\sin L & 0 & 0 \end{array}\right]$$
$$F_{22}=\left[\begin{array}{ccc} \frac{V_{n}\tan L}{R_{N}+H}-\frac{V_{u}}{R_{M}+H} & 2(\omega_{ie}\sin L+\frac{V_{e}\tan L}{R_{N}+H}) & -(2\omega_{ie}\cos L+\frac{V_{e}}{R_{N}+H})\\ -(2\omega_{ie}\sin L+\frac{V_{e}\tan L}{R_{N}+H}) & -\frac{V_{u}}{R_{M}+H} & -\frac{V_{n}}{R_{M}+H}\\ 2(\omega_{ie}\cos L+\frac{V_{e}}{R_{N}+H}) & \frac{2V_{n}}{R_{M}+H} & 0 \end{array}\right]$$
$$F_{23}=\left[\begin{array}{ccc} 0 & -f_{u} & f_{n}\\ f_{u} & 0 & -f_{e}\\ -f_{n} & f_{e} & 0 \end{array}\right]$$
$$F_{31}=\left[\begin{array}{ccc} 0 & 0 & 0\\ \frac{V_{e}\sec L\tan L}{R_{N}+H} & 0 & 0\\ 0 & 0 & 0 \end{array}\right]$$
$$F_{32}=\left[\begin{array}{ccc} 0 & \frac{1}{R_{M}+H} & 0\\ \frac{\sec L}{R_{N}+H} & 0 & 0\\ 0 & 0 & 1 \end{array}\right]$$
$$F_{44}=diag\left[\begin{array}{ccc} -\frac{1}{\tau_{\nabla_{x}}} & -\frac{1}{\tau_{\nabla_{y}}} & -\frac{1}{\tau_{\nabla_{z}}} \end{array}\right]$$
$$F_{55}=diag\left[\begin{array}{ccc} -\frac{1}{\tau_{\varepsilon_{x}}} & -\frac{1}{\tau_{\varepsilon_{y}}} & -\frac{1}{\tau_{\varepsilon_{z}}} \end{array}\right]$$
where $X_{k}$ is the system state vector; $\Phi_{k/k-1}$ is the state transition matrix; $\omega_{ie}$ is the Earth’s rotation angular velocity; L and H are the geographic latitude and altitude, respectively; $\left(V_{e},V_{n},V_{u}\right)$ are the velocities of the carrier in the navigation coordinate system; $R_{M}$ and $R_{N}$ are the curvature radii of the prime and meridian circles at the location of the carrier; $\left(f_{e},f_{n},f_{u}\right)$ are the specific force outputs of the accelerometer; $C_{b}^{n}$ is the coordinate conversion matrix from the navigation coordinate system to the carrier coordinate system; $F_{44}$ and $F_{55}$ are the zero-bias error state transition matrices of accelerometers and gyroscopes based on first-order Gaussian Markov processes; and $t_{\alpha }$ is the clock drift correlation time coefficients based on first-order Gaussian Markov processes. $w_{k-1}$ is the process noise vector, and its mean and covariance are as follows: (4)$$E[w_{k}]=0$$
(5)$$E[w_{k},w_{i}^{T}]=\{\begin{array}{c} Q_{k}i=k\\ 0i\neq k \end{array}$$
where E[·] is the mean and $Q_{k}$ is the process noise covariance matrix.

Generally, the measurement model established based on the difference between the pseudorange, carrier phase, and Doppler measurement values of the GNSS and INS are as follows: (6)$$Z_{k}=\left[\begin{array}{c} \nabla \Delta P_{GNSS}-\nabla \Delta P_{INS}\\ \nabla \Delta L_{GNSS}-\nabla \Delta L_{INS}\\ \nabla \Delta D_{GNSS}-\nabla \Delta D_{INS} \end{array}\right]=H_{k}X_{k}+v_{k}$$
(7)$$H_{k}=\left[\begin{array}{ccccccc} D\cdot C & 0_{n\times 3} & 0_{n\times 3} & 0_{n\times 6} & 0_{n\times 1} & 0_{n\times 1} & 0_{n\times n}\\ 0_{n\times 3} & E\cdot C & 0_{n\times 3} & 0_{n\times 6} & 0_{n\times 1} & 0_{n\times 1} & 0_{n\times n}\\ D\cdot C & 0_{n\times 3} & 0_{n\times 3} & 0_{n\times 6} & 0_{n\times 1} & 0_{n\times 1} & \lambda {Ι}_{n\times n} \end{array}\right]$$
where $Z_{k}$ represents the measurement vector; $H_{k}$ is the measurement matrix; D and E represent the design matrices of position error vectors and velocity error vectors, respectively; C represents the error conversion matrix between the spatial Cartesian coordinate system and the geodetic coordinate system; $v_{k}$ is the measurement noise vector; $v_{k}$ and $w_{k-1}$ are uncorrelated white noise sequences, and their mean and covariances are as follows:(8)$$E[w_{k}]=0$$
(9)$$E[w_{k},v_{{}_{k}}^{T}]=0$$
(10)$$E[v_{k},v_{i}^{T}]=\{\begin{array}{c} R_{k}i=k\\ 0i\neq k \end{array}$$
where E[·] is the mean, and $R_{k}$ is the measurement noise covariance matrix. In addition, owing to the short baseline differential positioning model used in this paper, errors such as ionospheric delay and tropospheric delay can be reduced or eliminated.

### 2.2. Extended Kalman Filter

Generally, a Kalman filter includes two parts, a state update and a measurement update, and a linear mathematical model is required. Unfortunately, in practical applications, most systems such as the measurement model in GNSS/INS tightly integrated navigation are nonlinear. In this case, the nonlinear equation needs to be linearized, and the EKF is a widely used method that involves Taylor expansion of nonlinear functions. 

State update of EKF:(11)$$X_{k/k-1}=\Phi_{k/k-1}X_{k-1}$$
(12)$$P_{k/k-1}=\Phi_{k/k-1}P_{k-1}\Phi_{k/k-1}^{T}+Q_{k-1}$$
where $X_{k/k-1}$ and $P_{k/k-1}$ are the predicted state vector and its covariance; $X_{k-1}$ and $P_{k-1}$ represent the state update vector and its covariance.

Measurement update steps of EKF:(a)The gain matrix ***K**_k_*:
(13)$$K_{k}=P_{k/k-1}H_{k}^{T}{\left(H_{k}P_{k/k-1}H_{k}^{T}+R_{k}\right)}^{-1}$$(b)The innovation vector ***V**_k_*:
(14)$$V_{k}=Z_{k}-H_{k}X_{k/k-1}$$(c)The state update vector ***X**_k_*:
(15)$$X_{k}=X_{k/k-1}+K_{k}V_{k}$$(d)The covariance of state update vector ***P**_k_*:
(16)$$P_{k}=(I-K_{k}H_{k})P_{k/k-1}$$
where ***K**_k_* describes the weights between new measurements and predicted information in the system dynamics model.

## 3. ZUPT-Based GNSS/IMU Tightly Coupled Algorithm with the Constraint of the Earth’s Rotation Angular Velocity

In this section, in order to obtain reliable position and attitude results using the GNSS/IMU tightly coupled algorithm, the zero-velocity update was added to the measurement model, and the constraint of the Earth’s angular velocity was added to the state model.

### 3.1. ZUPT

When the carrier is in a stationary state, the velocity of the carrier is zero. Inertial navigation errors can be corrected through IMU zero-velocity detection technology [41]. ZUPT utilizes the calculated velocity of the IMU in the carrier as the observation of the system velocity error, corrects other error quantities, and improves the integrated navigation results in a stationary state without the need for external sensors. Therefore, it is an effective, inexpensive, and easy-to-implement technology. Therefore, when the carrier is judged to be in a zero-velocity state, the zero-velocity constraint equation will be added to the observation Equation (5) to form the zero-velocity observation equation: (17)$$Z_{k}^{ZUPT}=\left[\begin{array}{c} \nabla \Delta P_{GNSS}-\nabla \Delta P_{INS}\\ \nabla \Delta L_{GNSS}-\nabla \Delta L_{INS}\\ \nabla \Delta D_{GNSS}-\nabla \Delta D_{INS}\\ V_{INS}-0_{3\times 1} \end{array}\right]=H_{k}^{ZUPT}X_{k}+v_{k}$$
(18)$$H_{k}^{ZUPT}=\left[\begin{array}{ccccccc} D\cdot C & 0_{n\times 3} & 0_{n\times 3} & 0_{n\times 6} & 0_{n\times 1} & 0_{n\times 1} & 0_{n\times n}\\ 0_{n\times 3} & E\cdot C & 0_{n\times 3} & 0_{n\times 6} & 0_{n\times 1} & 0_{n\times 1} & 0_{n\times n}\\ D\cdot C & 0_{n\times 3} & 0_{n\times 3} & 0_{n\times 6} & 0_{n\times 1} & 0_{n\times 1} & \lambda {Ι}_{n\times n}\\ 0_{3\times 3} & {Ι}_{3\times 3} & 0_{3\times 3} & 0_{3\times 6} & 0_{n\times 1} & 0_{n\times 1} & 0_{n\times n} \end{array}\right]$$

The updated state estimation value is used to provide feedback to and correct the navigation parameter error inside the IMU, completing the static zero-velocity correction. The precision of the zero-velocity constraint equation is extremely high. By giving a small prior variance, the Kalman filtering parameter of the zero-velocity additional equation is used to solve for strong constraints, achieving the goal of correcting the accumulation of static state errors.

In the static deformation monitoring environment of the cableway, the constraint method of the zero-velocity update (ZUPT) is particularly useful for the static GNSS/IMU tightly coupled deformation monitoring data. It was found that by constraining the three-axis velocity of inertial navigation, the divergence of the Kalman filter can be effectively suppressed and the corresponding pose data can be successfully calculated. However, this technology cannot suppress the drift of the combined positioning and attitude error.

### 3.2. Constraint of the Earth’s Rotation Angular Velocity

Although ZUPT has a certain inhibitory effect on the Kalman filter divergence when the carrier is stationary, it is just the constraint on the measurement model. There may still be position divergence and yaw drift during the calculation process, because the rotation of the navigation system caused by the curvature of the Earth’s surface in the state of the velocity update in the inertial navigation mechanical arrangement:(19)$$\omega_{ie}^{n}=C_{e}^{n}\omega_{ie}={\left[\begin{array}{ccc} 0 & \omega_{ie}\cos L & \omega_{ie}\sin L \end{array}\right]}^{T}$$
(20)$$\omega_{en}^{n}={\left[\begin{array}{ccc} -\frac{V_{n}}{R_{M}+H} & \frac{V_{e}}{R_{N}+H} & \frac{V_{e}}{R_{N}+H}\tan L \end{array}\right]}^{T}$$
(21)$$\omega_{in}^{n}=\omega_{ie}^{n}+\omega_{en}^{n}$$
where $\omega_{ie}^{n}$ is the angular velocity; the subscript ie represents the direction of rotation (the rotation of the Earth-fixed coordinate system relative to the inertial coordinate system); the superscript n represents the corresponding coordinate system (which, here, is the navigation coordinate system); $C_{e}^{n}$ is the coordinate conversion matrix from the navigation coordinate system to the Earth-fixed coordinate system; and $\omega_{ie}$ is the Earth’s rotation angular velocity.

To address the problem mentioned above, based on the carrier being stationary, a further constraint on the Earth’s angular velocity is applied to the state model under the ZUPT-based tightly coupled solution, that is, to constrain the yaw direction component of the Earth’s rotation angular velocity by setting: (22)$$\frac{V_{e}}{R_{N}+H}=0$$

As a result, the state transition matrix (3) with the constraint of the Earth’s rotation angular velocity (CERAV) is: (23)$$\Phi_{k/k-1}^{CERAV}=\left[\begin{array}{cccccccc} F_{11} & F_{12} & F_{13} & 0 & -C_{b}^{n} & 0 & 0 & 0\\ F_{21} & F_{22} & F_{23} & C_{b}^{n} & 0 & 0 & 0 & 0\\ 0 & F_{32} & 0 & 0 & 0 & 0 & 0 & 0\\ 0 & 0 & 0 & F_{44} & 0 & 0 & 0 & 0\\ 0 & 0 & 0 & 0 & F_{55} & 0 & 0 & 0\\ 0 & 0 & 0 & 0 & 0 & 0 & 1 & 0\\ 0 & 0 & 0 & 0 & 0 & 0 & -t_{\alpha } & 0\\ 0 & 0 & 0 & 0 & 0 & 0 & 0 & 0 \end{array}\right]$$
$$F_{11}=\left[\begin{array}{ccc} 0 & 0 & 0\\ -\omega_{ie}\sin L & 0 & 0\\ \omega_{ie}\cos L & 0 & 0 \end{array}\right]$$
$$F_{13}=\left[\begin{array}{ccc} 0 & \omega_{ie}\sin L & -\omega_{ie}\cos L\\ -\omega_{ie}\sin L & 0 & -\frac{V_{n}}{R_{M}+H}\\ \omega_{ie}\cos L & \frac{V_{n}}{R_{M}+H} & 0 \end{array}\right]$$
$$F_{21}=\left[\begin{array}{ccc} 2\omega_{ie}V_{n}\cos L+2\omega_{ie}V_{u}\sin L & 0 & 0\\ -2\omega_{ie}V_{e}\cos L & 0 & 0\\ -2\omega_{ie}V_{e}\sin L & 0 & 0 \end{array}\right]$$
$$F_{22}=\left[\begin{array}{ccc} \frac{V_{n}\tan L}{R_{N}+H}-\frac{V_{u}}{R_{M}+H} & 2\omega_{ie}\sin L & -2\omega_{ie}\cos L\\ -2\omega_{ie}\sin L & -\frac{V_{u}}{R_{M}+H} & -\frac{V_{n}}{R_{M}+H}\\ 2\omega_{ie}\cos L & \frac{2V_{n}}{R_{M}+H} & 0 \end{array}\right]$$

### 3.3. ZUPT-Based GNSS/IMU Tightly Coupled Algorithm with CERAV

According to the descriptions in Section 3.1 and Section 3.2, the flowchart for practical implementation of the ZUPT-based GNSS/IMU tightly coupled algorithm with CERAV is shown in Figure 1.

## 4. Experiments and Results Analysis

In this section, the proposed method, called the ZUPT-based GNSS/IMU tightly coupled algorithm with CERAV (ZUPT-TC-CERAV), is compared with the ZUPT-based GNSS/IMU tightly coupled algorithm (ZUPT-TC). Due to the divergence of the Kalman filter, the GNSS/IMU tightly coupled algorithm without constraint cannot provide position and attitude solutions, so it is not considered. In addition, the GNSS alone (which can only provide the position not affected by IMU divergence, but cannot provide attitude) is used to demonstrate the effectiveness of the proposed method. 

Experimental data were collected on the crossbar of the cableway in the Yimeng Mountain Tourism area on 23 September 2022, over a duration of approximately 15 min. The environment of the cableway bracket is shown in Figure 2. The tourist sightseeing cable car station is located in the southern area of the scenic area. The pick-up station is located at the top of the main peak tower in the southern area, and the drop-off station is located at Waterfall Square. The one-way passenger capacity of the sightseeing cable car is 1668 people/h, and each carriage can accommodate 160 people. The inclined length of the route is 1835.5 m.

The experimental platform consists of two Hi-Target receivers and a SPAN-IGM-A1 IMU. The update rate of the Hi-Target receiver is 1 Hz, and that of the SPAN-IGM-A1 IMU is 200 Hz. The technical parameters for the SPAN-IGM-A1 IMU are listed in Table 1, and the experimental platform is shown in Figure 3.

Figure 4 is a schematic diagram of the position relationship between the navigation coordinate and the cableway coordinate. In this paper, the baseline has been transformed from the navigation coordinate (E-N-U) to the cableway coordinate (X-Y-Z) in order to facilitate the exploration of position and attitude. The number of satellites and dilution of precision (DOP) are shown in Figure 5 and Figure 6. It is obvious that the satellite observation environment is good.

Figure 7 and Figure 8, respectively, contain the position of GNSS, ZUPT-TC, and ZUPT-TC-CERAV in navigation coordinates and cableway coordinates, and it can be seen that the position solution can be obtained by applying ZUPT, but the problem of filtering divergence still exists. Especially after turning the coordinates, the position of ZUPT-TC diverged in all three directions. Meanwhile, the position of ZUPT-TC-CERAV is relatively stable owing to the constraint of the state model, which also leads to the difference in the initial position of ZUPT-TC and ZUPT-TC-CERAV. In addition, the phenomenon that the position of ZUPT-TC-CERAV basically matches the position of the GNSS, which is not affected by IMU divergence, verifies the effectiveness of ZUPT-TC-CERAV again.

The attitude of ZUPT-TC and ZUPT-TC-CERAV are provided in Figure 9. Obviously, compared to ZUPT-TC-CERAV, the trends of roll and pitch obtained by ZUPT-TC are relatively stable, while that of yaw shows divergence. At the same time, the trends of roll, pitch, and yaw obtained by ZUPT-TC-CERAV are all stable. The reason for this could be that the drifting error of yaw in the stationary state is significantly suppressed by adding the constraint of Earth’s rotation angular velocity.

To further illustrate the difference between ZUPT-TC and ZUPT-TC-CERAV, the position error and attitude error are shown in Figure 10 and Figure 11. It is indicated that the position error of ZUPT-TC shows a divergent trend in the X, Y, and Z directions, while the errors of the ZUPT-TC-CERAV solution are all relatively stable, and the difference in the position error between GNSS and ZUPT-TC-CERAV is small. Similarly, the same divergent trend caused by the stationary IMU is shown in the yaw error of ZUPT-TC, and ZUPT-TC-CERAV can effectively solve this problem.

The RMSs of the position and attitude errors of ZUPT-TC and ZUPT-TC-CERAV are shown in Table 2. It is indicated that (a) the accuracies of the position errors obtained by ZUPT-TC-CERAV are improved by 92.41%, 91.02%, and 68.59% in the X, Y, and Z directions, respectively, compared to ZUPT-TC and (b) compared with ZUPT-TC, the RMS of the yaw error obtained by ZUPT-TC-CERAV shows that performance improved by 98.77%.

From the description in Table 2 and the analysis of the experimental results, we can observe that ZUPT-TC-CERAV is able to overcome the problem of Kalman filter divergence and drifting error caused by the static IMU measurement, in contrast to ZUPT-TC, and its reliable performance of positioning and attitude determination is demonstrated by the minor difference from single GNSS.

## 5. Discussion

The cableway bracket used to support the normal operation of the steel wire rope plays an important role in the detection of potential faults and abnormal states, as the cableway will have a significant impact on the three-dimensional deformation and attitude changes in the cableway bracket when an accident occurs. Although a single GNSS algorithm can provide a certain degree of accuracy in position information for deformation monitoring, it cannot provide attitude changes. Therefore, the GNSS/IMU tightly coupled algorithm is a must to obtain high-precision position and attitude data for cableway bracket deformation monitoring. However, when the IMU is stationary for a long time, it can cause significant drift errors. As a result, the GNSS/IMU tightly coupled algorithm without constraint cannot provide position and attitude solutions.

To overcome this problem, the ZUPT and the constraint of the Earth’s rotation angular velocity are applied to the measurement model and the state model of the Kalman filter. By adding the zero-velocity constraint equation to the observation equation, ZUPT-TC can obtain position and attitude solutions, but there will still be drift in the position and yaw. Furthermore, the constraint of the Earth’s rotation angular velocity is added to the state model, and then the position and attitude that match the actual situation are obtained. However, the absolute accuracy of the proposed method is still unavailable (because the reliable solution cannot be obtained), so we can only obtain its relative accuracy by comparing its solution with that of single GNSS.

## 6. Conclusions

In this paper, a ZUPT-based GNSS/IMU tightly coupled algorithm with the constraint of the Earth’s rotation angular velocity, which effectively solves the problem of Kalman filter divergence and drifting error caused by static measurement, is proposed for cableway bracket deformation monitoring. A performance assessment of the proposed scheme was conducted in the Yimeng Mountain Tourism area. It was demonstrated that the proposed scheme is superior to the ZUPT-based algorithm, especially in position and yaw, which are important to the monitoring of cableway bracket deformation.

## Figures and Tables

**Figure 1 sensors-23-09862-f001:**
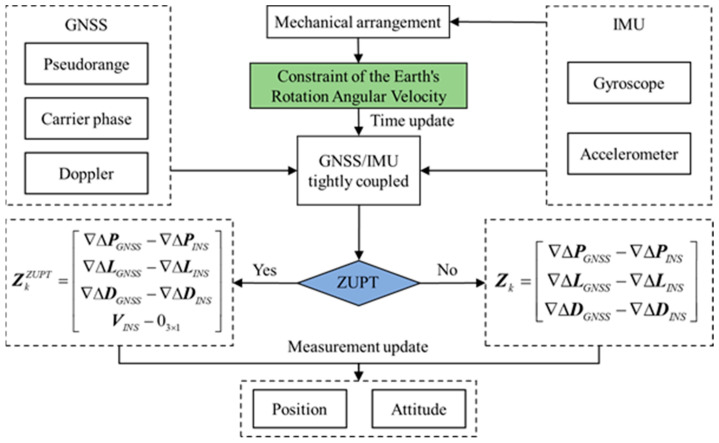
The flowchart of the ZUPT-based GNSS/IMU tightly coupled algorithm with CERAV.

**Figure 2 sensors-23-09862-f002:**
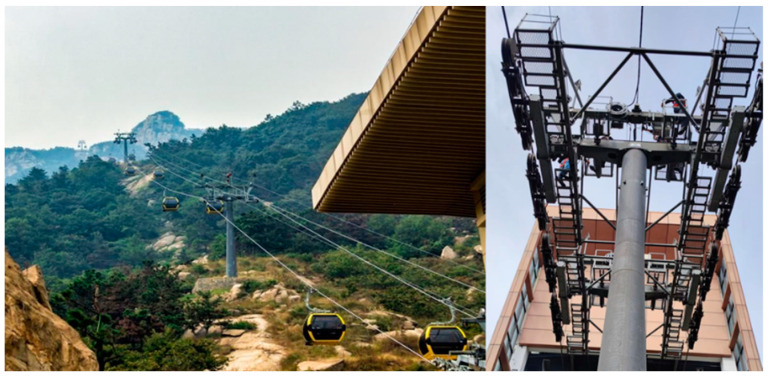
The environment of cableway bracket.

**Figure 3 sensors-23-09862-f003:**
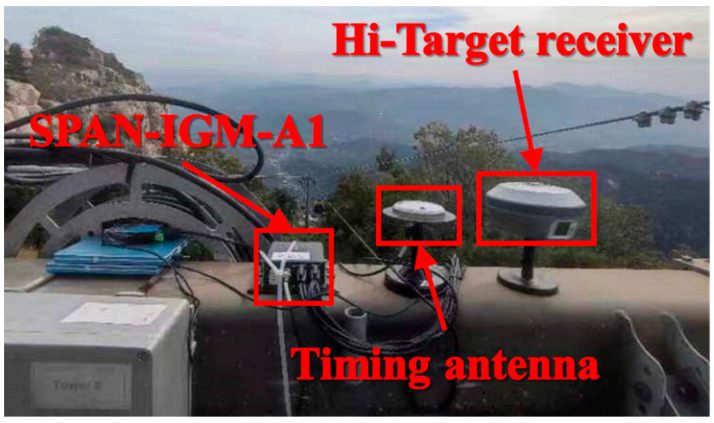
Experimental platform used in the test.

**Figure 4 sensors-23-09862-f004:**
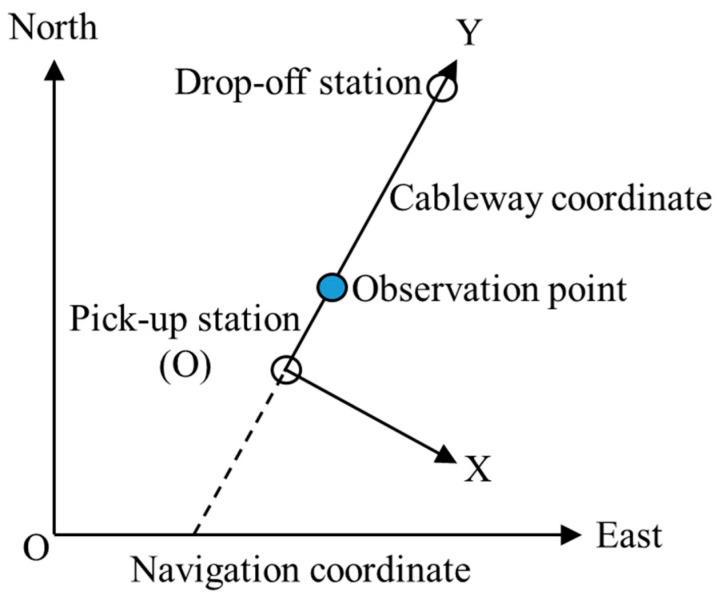
Position relationship between navigation coordinate and cableway coordinate.

**Figure 5 sensors-23-09862-f005:**
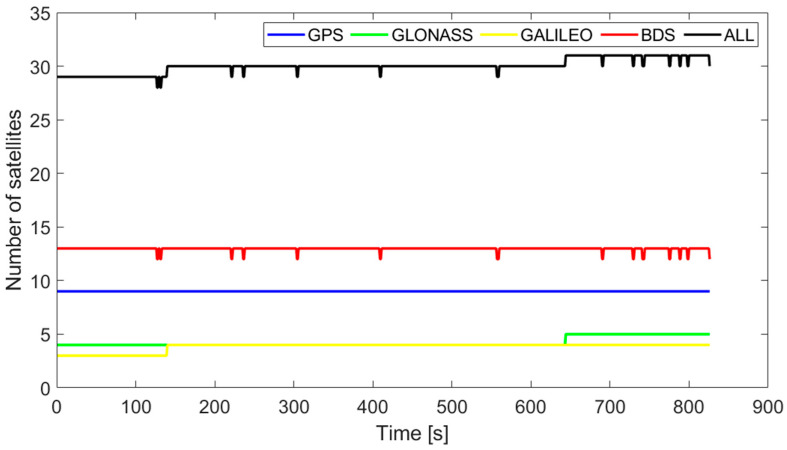
The number of satellites.

**Figure 6 sensors-23-09862-f006:**
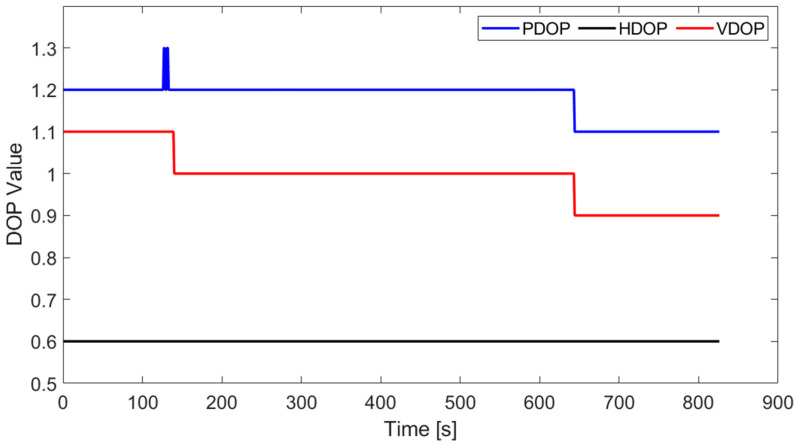
DOP value.

**Figure 7 sensors-23-09862-f007:**
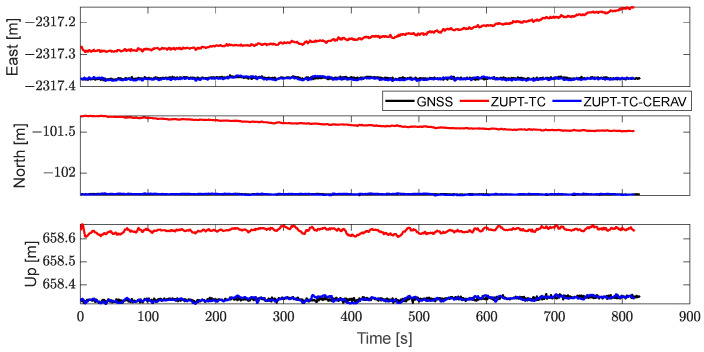
The position of GNSS, ZUPT-TC, and ZUPT-TC-CERAV in navigation coordinate.

**Figure 8 sensors-23-09862-f008:**
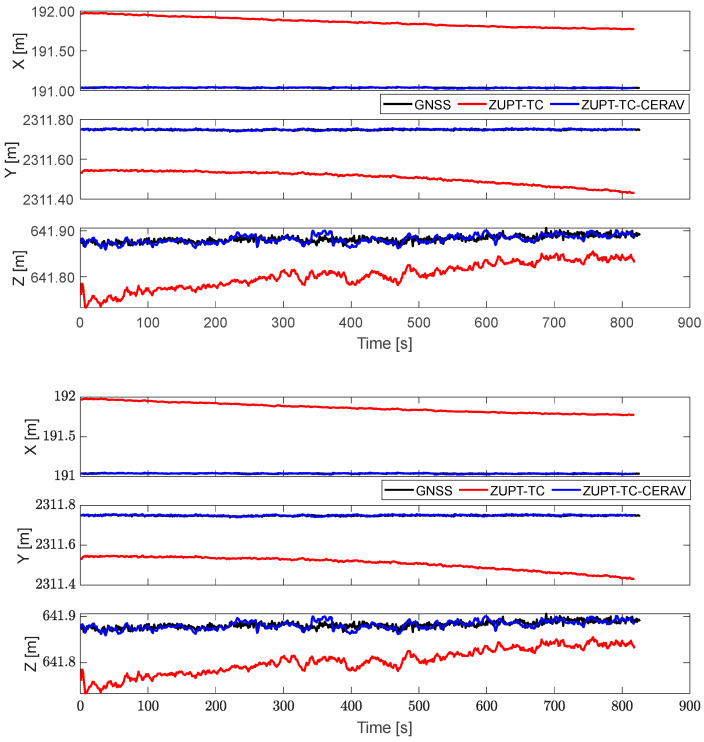
The position of GNSS, ZUPT-TC, and ZUPT-TC-CERAV in cableway coordinate.

**Figure 9 sensors-23-09862-f009:**
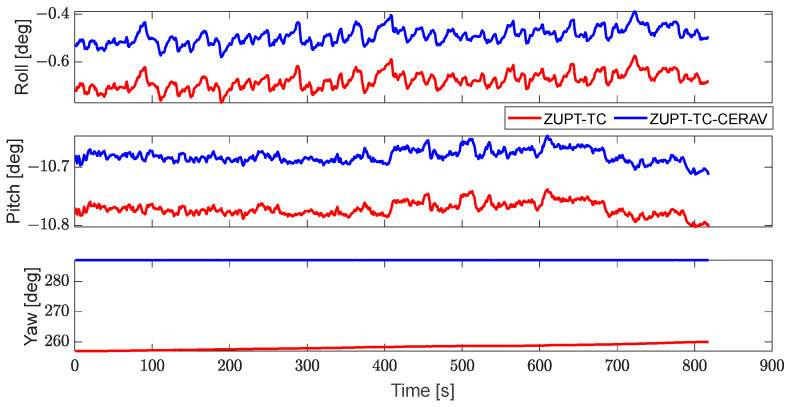
The attitudes of ZUPT-TC and ZUPT-TC-CERAV.

**Figure 10 sensors-23-09862-f010:**
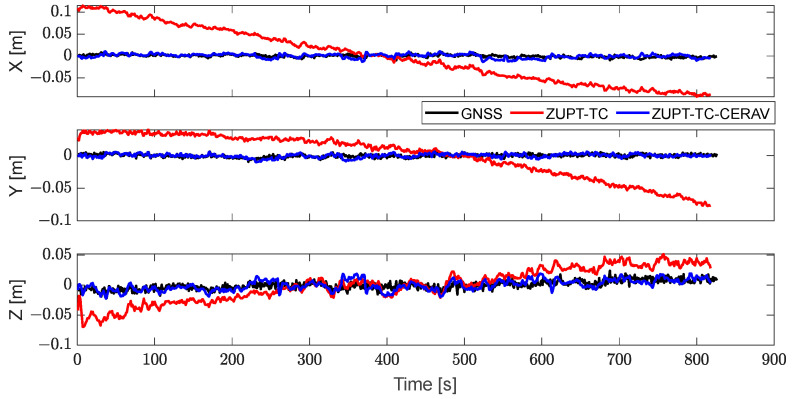
The position errors of GNSS, ZUPT-TC and ZUPT-TC-CERAV.

**Figure 11 sensors-23-09862-f011:**
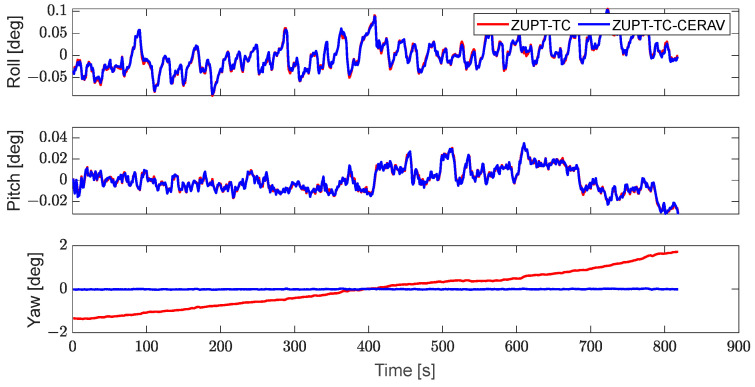
The attitude errors of ZUPT-TC and ZUPT-TC-CERAV.

**Table 1 sensors-23-09862-t001:** SPAN-IGM-A1 technical parameters.

IMU	Bias Instability	Random Walk
Accelerometer	0.1 mg	0.029 m/s/sqrt(h)
Gyroscope	6 deg/h	0.3 deg/sqrt(h)

**Table 2 sensors-23-09862-t002:** The RMSs of position and attitude errors of ZUPT-TC and ZUPT-TC-CERAV.

	Position (m)			Attitude (deg)		
	X	Y	Z	Roll	Pitch	Yaw
ZUPT-TC	0.0619	0.3340	0.0277	0.0331	0.0114	0.8423
ZUPT-TC-CERAV	0.0047	0.0030	0.0087	0.0326	0.0114	0.0104

## Data Availability

The data that support the findings of this study are available from the corresponding author upon reasonable request.

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
