# Peer review of "Zero-Velocity Update-Based GNSS/IMU Tightly Coupled Algorithm with the Constraint of the Earth’s Rotation Angular Velocity for Cableway Bracket Deformation Monitoring"

_sensors, 2023, doi:10.3390/s23249862_

Round 1

Reviewer 1 Report

Comments and Suggestions for Authors

The paper by Song Zhang, Qiuzhao Zhang et al. ZUPT-Based GNSS/IMU Tightly Coupled Algorithm with the Constraint of the Earth's Rotation Angular Velocity for Cable-Way Bracket Deformation Monitoring

covers the process of estimating the state vector of the cable car cabin using GNSS.

The authors developed the method of correction of the drift of coordinates and attitude (Figs. 10-11) using the assumption, that the eastward projection of the Earth rotation angular velocity should be zero. Generally speaking, I do not see any arguments why not to publish their research, but to make it more scientifically sound, I would ask the following questions: I might not be quite familiar with the ZUPT algorithm, but in my opinion, it was developed by qualified specialists and is based on real geometry of GNSS tracking.

The authors decided to make the term Ve/(Rn+H) equal zero everywhere in the design matrix. And as I understood, the drift in the positions of the cabin at rest disappeared. I would call this not an extension but a simplification of the algorithm scheme. I understand that only Ve divided by a large number (Rn+H) was made equal zero. It remains where it stands, multiplied by sinL or cosL in the matrices. But are the authors sure that their algorithm would not give wrong estimates in motion, or, let’s say, at another position on the Earth? I mean for cable cars in the countries, different from Shandong Province. Does their algorithm have more sound physical justification than just one particular, specific experiment? We even do not know if the drift of 10–20 cm was related to changes in ionosphere conditions, retroreflections, or any other circumstances. The explanation of the experiment is quite short.

As a specialist on Earth rotation, I would bring to the attention of the authors that the angular velocity vector of the Earth does have an eastward and Greenwich meridional component of projection, related to the motion of the pole. Of course, polar motion is at a level of 6 meters and cannot change in 15 minutes, but it exists. Thus, the assumption of zero eastward projection Ve does not, in reality, use any apriori information but just makes zero particular terms in the expressions. It is not quite a constraint, which is based on real additional knowledge of what is happening, but a convenient way of neglecting the term, which is not quite important for a particular problem. Thus, I would not make the name of the article so sound in terms of Earth rotation velocity. I would just say: algorithm with the assumption that the eastward projection of Earth's angular velocity is equal to zero, or something like this.

So, my main issue is: could you solve the problem by understanding the real reasons for the drift and incorporating this into the procedure, not just by dropping some, which might be small, but physical terms in the equations?

Some smaller issues:  is it necessary to repeat the matrices that have not changed in 3.2?

What is the dimensionality of eq. 18, does it agree with state–vector?

Is double difference ambiguity also included into observation scheme? Is the system observable?

Do you think it is correct to compare the solution with drift with the solution without in terms of STD? I mean, saying that the new solution is 99% better, just because the drif variance is much larger than the other superimposed oscillations. 

Comments on the Quality of English Language

English is fine

Author Response

Response to Reviewer 1 Comments

Dear Editors and Reviewers:

Thank you for your letter and for the reviewers’ comments concerning our manuscript ID sensors-2680390 entitled " ZUPT-Based GNSS/IMU Tightly Coupled Algorithm with the Constraint of the Earth's Rotation Angular Velocity for Cableway Bracket Deformation Monitoring". Those comments are all valuable and very helpful for revising and improving our paper, as well as the important guiding significance to our research. Revised portions are marked in the manuscript. The main corrections revision in the paper and the response to the reviewer’s comments are as follows:

Comments and Suggestions for Authors

The authors developed the method of correction of the drift of coordinates and attitude (Figs. 10-11) using the assumption, that the eastward projection of the Earth rotation angular velocity should be zero. Generally speaking, I do not see any arguments why not to publish their research, but to make it more scientifically sound, I would ask the following questions: I might not be quite familiar with the ZUPT algorithm, but in my opinion, it was developed by qualified specialists and is based on real geometry of GNSS tracking.

Point 1: The authors decided to make the term Ve/(Rn+H) equal zero everywhere in the design matrix. And as I understood, the drift in the positions of the cabin at rest disappeared. I would call this not an extension but a simplification of the algorithm scheme. I understand that only Ve divided by a large number (Rn+H) was made equal zero. It remains where it stands, multiplied by sinL or cosL in the matrices. But are the authors sure that their algorithm would not give wrong estimates in motion, or, let’s say, at another position on the Earth? I mean for cable cars in the countries, different from Shandong Province. Does their algorithm have more sound physical justification than just one particular, specific experiment? We even do not know if the drift of 10–20 cm was related to changes in ionosphere conditions, retroreflections, or any other circumstances. The explanation of the experiment is quite short.

Response 1: Thank you for your comment.

   are the velocities of the carrier in navigation coordinate system (in a stationary state, it should be equal to zero). Based on the prior information that the carrier is almost stationary, ZUPT was used in the measurement model to restrain Kalman filter divergence. Unfortunately, the constraint of the measurement model is not enough, as Kalman filter includes both the state model and the measurement model. Therefore, to further constrain the attitude divergence, we make the term Ve/(Rn+H) (which is related to Earth's rotation angular velocity) equal zero (because Ve equal zero) everywhere in the design matrix. (Lines 228-242, page 7)

We are certain that the algorithm proposed in this manuscript will definitely give wrong estimates in motion, because the constraints are based on the condition that the experiment platform on the cableway bracket (not cable car) be monitored is stationary.

To provide more sound physical justification of the algorithm, in the revised manuscript, the position solution provided by single GNSS (which can only provide position, and cannot provide attitude) was added to compare with the positions of ZUPT-TC and ZUPT-TC-CERAV in navigation coordinate and cableway coordinate. It can be seen that the position error between GNSS and ZUPT-TC-CERAV is little, due to the Kalman filter divergence caused by IMU has been restrained successfully, and it was also proved that the drift of 10–20 cm caused by IMU was not related to changes in ionosphere conditions or retroreflections. Besides, owing to the short baseline differential positioning model used in this paper, errors such as ionospheric delay and tropospheric delay can be reduced or eliminated.

The position error of GNSS, ZUPT-TC and ZUPT-TC-CERAV in navigation coordinate

The position error of GNSS, ZUPT-TC and ZUPT-TC-CERAV in cableway coordinate

Point 2: As a specialist on Earth rotation, I would bring to the attention of the authors that the angular velocity vector of the Earth does have an eastward and Greenwich meridional component of projection, related to the motion of the pole. Of course, polar motion is at a level of 6 meters and cannot change in 15 minutes, but it exists. Thus, the assumption of zero eastward projection Ve does not, in reality, use any apriori information but just makes zero particular terms in the expressions. It is not quite a constraint, which is based on real additional knowledge of what is happening, but a convenient way of neglecting the term, which is not quite important for a particular problem. Thus, I would not make the name of the article so sound in terms of Earth rotation velocity. I would just say: algorithm with the assumption that the eastward projection of Earth's angular velocity is equal to zero, or something like this.

Response 2: Thank you for your comment.  are the velocities of the carrier in navigation coordinate system (in a stationary state, the velocities should be equal to zero). Based on the prior information that the carrier is stationary, to further constrain the attitude divergence from the state model, we make the term Ve/(Rn+H) (which is related to Earth's rotation angular velocity) equal zero (because Ve equal zero) everywhere in the design matrix. Therefore, we believe it is a constraint based on the real additional knowledge that the carrier is stationary.

Point 3: So, my main issue is: could you solve the problem by understanding the real reasons for the drift and incorporating this into the procedure, not just by dropping some, which might be small, but physical terms in the equations?

Response 3: Thank you for your suggestion. The real reasons for the drift are as follows:

(1) If only position was used for the cableway bracket deformation monitoring, we can use single GNSS, and then there won't be any drift. However, merely utilizing position for deformation monitoring is not enough, the three-dimensional attitude changes of the cableway bracket are equally important. It was demonstrated that GNSS/IMU combination can improve accuracy and robustness compared to single GNSS. Therefore, GNSS/IMU tightly coupled algorithm was used to provide the position and attitude of the cableway bracket.

(2) In a stationary state, the error of IMU will gradually accumulate over time, as a result, Kalman filtering will diverge. Hence, according to the stationary state, ZUPT was used in the measurement model, and the constraint of the Earth's rotation angular velocity was applied in the state model.

Point 4: is it necessary to repeat the matrices that have not changed in 3.2?

Response 4: Thank you for your suggestion. We have removed the matrices that have not changed in 3.2. (Lines 245-248, page 9)

Point 5: What is the dimensionality of eq. 18, does it agree with state–vector?

Response 5: Thank you for your suggestion. The columns of eq. 18 are agree with state–vector, and the rows of eq. 18 are agree with measurement–vector.

Point 6: Is double difference ambiguity also included into observation scheme? Is the system observable?

Response 6: Thank you for your comment. Double difference ambiguity is the unknown parameter contained in the state vector, so it is not included in the observation scheme. We can get it by the solution of Kalman filtering and the LAMBDA algorithm. Yes, the system is observable, GNSS/IMU tightly coupled algorithm will work with only at least three satellites, and in Section 5, the number of satellites is enough.

The number of satellites

Point 7: Do you think it is correct to compare the solution with drift with the solution without in terms of STD? I mean, saying that the new solution is 99% better, just because the drif variance is much larger than the other superimposed oscillations.

Response 7: Thank you for your comment. The purpose of the comparison in terms of STD is to verify the effectiveness of the constraints, because the solution of GNSS/IMU tightly coupled algorithm without the constraints cannot be obtained.

To further validate the performance of the algorithm, in the revised manuscript, we added the position solution provided by single GNSS to compare with the positions of ZUPT-TC and ZUPT-TC-CERAV in navigation coordinate and cableway coordinate. It can be seen that the position error between GNSS and ZUPT-TC-CERAV is small, which is consistent with the actual situation. Besides, owing to the short baseline differential positioning model used in this paper, errors such as ionospheric delay and tropospheric delay can be reduced or eliminated.

The position error of GNSS, ZUPT-TC and ZUPT-TC-CERAV in navigation coordinate

The position error of GNSS, ZUPT-TC and ZUPT-TC-CERAV in cableway coordinate

Reviewer 2 Report

Comments and Suggestions for Authors

The motivation is not well presented in the current manuscript. The formulation is hard to follow and discussions on the obtained results are missing, which weakened the strength of the study. Detailed comments are listed as follows:

(1) The literature review missed some important advancements in this area. The pointed existing issues in not so convincing. 

(2) Background should be included before formulation of the dynamic and observation equations.

(3) More current methods should be employed for comparison purpose.

(4) Discussion is missing in the current work.

(5) The writing should be improved and all figures should be presented in a uniform and clear manner. 

Comments on the Quality of English Language

The technical writing should be improved and all figures should be presented in a uniform and clear manner. 

Author Response

Response to Reviewer 2 Comments

Dear Editors and Reviewers:

Thank you for your letter and for the reviewers’ comments concerning our manuscript ID sensors-2680390 entitled " ZUPT-Based GNSS/IMU Tightly Coupled Algorithm with the Constraint of the Earth's Rotation Angular Velocity for Cableway Bracket Deformation Monitoring". Those comments are all valuable and very helpful for revising and improving our paper, as well as the important guiding significance to our research. Revised portions are marked in the manuscript. The main corrections revision in the paper and the responses to the reviewer’s comments are as follows:

Comments and Suggestions for Authors

The motivation is not well presented in the current manuscript. The formulation is hard to follow and discussions on the obtained results are missing, which weakened the strength of the study. Detailed comments are listed as follows:

Point 1: The literature review missed some important advancements in this area. The pointed existing issues in not so convincing.

Response 1: Thank you for your suggestion. We have added the important advancements in the revised manuscript to make the pointed existing issues more convincing. (Lines 33-45, pages 1-2; Lines 120-128, pages 3)

Point 2: Background should be included before formulation of the dynamic and observation equations.

Response 2: Thank you for your suggestion. The background has been included before the formulation of the dynamic and observation equations in the revised manuscript. (Lines 139-147, page 3)

Point 3: More current methods should be employed for comparison purpose.

Response 3: Thank you for your suggestion. In the revised manuscript, the position solution provided by single GNSS (which can only provide position, and cannot provide attitude) was added to compare with the positions of ZUPT-TC and ZUPT-TC-CERAV in navigation coordinate and cableway coordinate. It can be seen that the position error between GNSS and ZUPT-TC-CERAV is small, due to the Kalman filter divergence caused by IMU has been restrained successfully.

The position error of GNSS, ZUPT-TC and ZUPT-TC-CERAV in navigation coordinate

The position error of GNSS, ZUPT-TC and ZUPT-TC-CERAV in cableway coordinate

Point 4: Discussion is missing in the current work.

Response 4: Thank you for your suggestion. We have added the discussion in the revised manuscript. (Line 335-355, page 15)

Point 5: The writing should be improved and all figures should be presented in a uniform and clear manner.

Response 5: Thank you for your reminding. According to your suggestion, the writing of the manuscript has been thoroughly revised and edited, and all figures have been presented in a uniform and clear manner.

Round 2

Reviewer 1 Report

Comments and Suggestions for Authors

I am satisfied by the response of the authors to reviewer. Though I am still at the position, that the introduced constraint on the Earth rotation velocity does not in reality uses any new knowledge, but just drops some term to make the solution stable.

Being familiar with the inverse problems, I guess that estimation of both position and attitude makes the estimation scheme ill-posed with highly correlated coordinates of the state vector. In general, this problem can be solved by regularization or dropping of some most uncertainty state-vector directions. I risk to suppose, that CERAV here works as a kind of regularization. But nor Earth rotation velocity from measurements of the Earth rotation, neither any other corrections based on additional knowledge were not added as a constraint.

I do not think that my general assumptions could be a reason for stopping the publication of this paper with some new engineering achievements.

Reviewer 2 Report

Comments and Suggestions for Authors

The authors have addressed all my concerns.